# Tetrahydrocurcumin-Related Vascular Protection: An Overview of the Findings from Animal Disease Models

**DOI:** 10.3390/molecules27165100

**Published:** 2022-08-10

**Authors:** Li Zhang, Changhu Li, Sicheng Wang, Dimiter Avtanski, Nikola Hadzi-Petrushev, Vadim Mitrokhin, Mitko Mladenov, Feng Wang

**Affiliations:** 1Cancer Center, Department of Medical Oncology, West China Hospital, West China Medical School, Sichuan University, Chengdu 610041, China; 2Cancer Center, Division of Radiation Physics, West China Hospital, Sichuan University, Chengdu 610041, China; 3Medical Department, 6th City Clinical Hospital, 220037 Minsk, Belarus; 4Friedman Diabetes Institute, Lenox Hill Hospital, Northwell Health, 110 E 59th Street, New York, NY 10022, USA; 5Faculty of Natural Sciences and Mathematics, Institute of Biology, “Ss. Cyril and Methodius” University, P.O. Box 162, 1000 Skopje, North Macedonia; 6Department of Physiology, Pirogov Russian National Research Medical University, Ostrovityanova Street, 1, 117997 Moscow, Russia

**Keywords:** tetrahydrocurcumin, vasculature, endothelial cells, mitochondria, reactive oxygen species, antioxidants

## Abstract

Tetrahydrocurcumin (THC), one of the major metabolites of CUR, possesses several CUR-like pharmacological effects; however, its mechanisms of action are largely unknown. This manuscript aims to summarize the literature on the preventive role of THC on vascular dysfunction and the development of hypertension by exploring the effects of THC on hemodynamic status, aortic elasticity, and oxidative stress in vasculature in different animal models. We review the protective effects of THC against hypertension induced by heavy metals (cadmium and iron), as well as its impact on arterial stiffness and vascular remodeling. The effects of THC on angiogenesis in CaSki xenografted mice and the expression of vascular endothelial growth factor (VEGF) are well documented. On the other hand, as an anti-inflammatory and antioxidant compound, THC is involved in enhancing homocysteine-induced mitochondrial remodeling in brain endothelial cells. The experimental evidence regarding the mechanism of mitochondrial dysfunction during cerebral ischemic/reperfusion injury and the therapeutic potential of THC to alleviate mitochondrial cerebral dysmorphic dysfunction patterns is also scrutinized and explored. Overall, the studies on different animal models of disease suggest that THC can be used as a dietary supplement to protect against cardiovascular changes caused by various factors (such as heavy metal overload, oxidative stress, and carcinogenesis). Additionally, the reviewed literature data seem to confirm THC’s potential to improve mitochondrial dysfunction in cerebral vasculature during ischemic stroke through epigenetic mechanisms. We suggest that further preclinical studies should be implemented to demonstrate THC’s vascular-protective, antiangiogenic, and anti-tumorigenic effects in humans. Applying the methods used in the presently reviewed studies would be useful and will help define the doses and methods of THC administration in various disease settings.

## 1. Background

Curcumin (CUR) and its metabolites are well-researched natural compounds with numerous beneficial biological activities. Curcumin is produced by the plant *Curcuma longa*, and tetrahydrocurcumin (THC), as one of its primary in vivo metabolites, possesses several CUR-like pharmacological effects [1].

Multiple studies have investigated the protective effects of CUR and its metabolites/analogs on cardiovascular diseases (CVDs). Recent reviews of the accumulated data indicate that CUR is likely a therapeutic agent for CVDs. It exerts its pleiotropic actions by modulating various signaling pathways involved in oxidative stress, inflammation, glucose metabolism, mitochondrial function, apoptosis, and proliferation [2,3,4,5]. In addition, the finding that the disruption of vascular homeostasis is an early stage of many cardiovascular disorders has incited further interest in the beneficial effects of CUR and its metabolites on the vasculature [5,6,7].

Comparing the structure–activity relationships of CUR and THC revealed the basis for their shared beneficial bioactivities and emphasized THC’s improved stability [8]. Previous reviews have mainly focused on THC’s anti-inflammatory and anticancer effects [7,9]. However, although many experimental studies have indicated THC’s cardiovascular protective properties, to date, there has been only one CVD-themed review regarding the beneficial effects of THC (and CUR) in cardiovascular dysfunction associated with cadmium (Cd^2+^) exposure [10]. Since the accumulated experimental data hold promise for THC to be a potential therapeutic agent against CVDs, in a similar manner as CUR, the following review aims to specifically summarize these findings, with particular attention being given to the molecular mechanisms and signaling targets.

### 1.1. CUR- and THC-Related Vascular Protection in L-NAME-Induced Hypertension

Recent studies have shown that treatment with curcumin (CUR) or tetrahydrocurcumin (THC), particularly at high doses, restores the expression of endothelial nitric oxide synthase (eNOS) and prolongs its reactive oxygen species (ROS)- balancing action. According to Nakmareong et al. [11], CUR efficiently restores eNOS levels while decreasing the overproduction of superoxide (O_2_^−^) [12]. The same authors [12] found that CUR and THC cause a decrease in the progression of (ω)-nitro-L-arginine methyl ester (*L-NAME*)-induced hypertension without altering the hemodynamics in the normotensive rats. These findings suggest that CUR and THC are not hypotensive drugs and only have antihypertensive effects under pathological settings (especially in high doses) (Figure 1). De Gennaro Colonna et al. [13] reported that in response to acetylcholine (Ach) and angiotensin II (Ang II), CUR and THC restored the vascular responses to near-normal levels. More specifically, they found that CUR administration for six weeks improved the vascular response to Ach in diabetic rats with streptozotocin-induced endothelial dysfunction and that this effect was linked to reduced O_2_^−^ generation in the mesenteric arteries. Additionally, different CUR analogs protect the heart from ischemic damage and improve cardiac function by strengthening the antioxidant defense mechanisms [14,15,16,17]. According to Nakmareong et al. [11], the potential of CUR and THC to enhance the hemodynamic condition and restore aortic responsiveness is linked to the restoration of eNOS expression and blood plasma nitrate/nitrite levels. One of the reasons for these changes might be due to CUR’s or THC’s capacity to boost nitric oxide (NO) production, which restricts the production of oxidants and improves NO bioavailability to a level adequate for “normal” vascular function.

Several polyphenolic compounds isolated from the turmeric plant (*Curcuma longa*), which is the source of the dietary spice turmeric, have been demonstrated to exhibit significant antioxidant action through decreasing ROS generation by direct inhibition of ROS-generating enzymes. Xanthine oxidase and lipoxygenase are two such examples (Figure 1). CUR, on the other hand, has been demonstrated to activate nuclear factor erythroid-derived 2-like 2 (Nrf2), resulting in the increased expression of antioxidant enzyme genes, such as superoxide dismutase (SOD), catalase (CAT), glutamate-cysteine ligase (GCL), and NAD(P)H oxidoreductase [18,19] (Figure 1). When L-NAME hypertensive mice were treated with CUR or THC, oxidant production was considerably decreased by a reduction in the levels of O_2_^−^, malondialdehyde (MDA), and protein carbonyls, while endogenous antioxidant glutathione (GSH) was boosted by raising the GSH ratio in the redox direction (Figure 1). THC, on the other hand, does not appear to be able to activate Nrf2 [20], and its role in the induction of antioxidant enzyme expression is currently unclear.

The mechanisms of action of CUR and THC are based on the fact that both compounds have similar β-diketone structures and phenolic groups; however, THC lacks CUR’s double bonds in the seventh carbon linker of the molecule [21]. Although CUR and THC have identical active forms of β-diketone structures and phenolic groups [22], THC is more soluble in aqueous solutions and easier to be absorbed via the gastrointestinal tract [21]. This might explain why THC has better antioxidant and antihypertensive properties than CUR. Although THC is a key metabolite, the quantity of THC generated in vivo from CUR is modest [23], and thus it could be expected to have a reducing role in CUR’s activity.

### 1.2. THC-Induced Vascular Protection in the Context of NO Deficit

THC has been proven to have anti-inflammatory properties via decreasing IkappaB kinase (IKK) activity and nuclear factor-kappa B (NF-κB) activation [24]. According to Kitamoto et al. [25], the long-term inhibition of NOS caused increased NF-κB and activator protein 1 (AP-1) activities in the aortic walls of rats [25]. The inhibition of the NF-κB activity is expected to decrease the expression of its target genes, most notably NADPH oxidase [26,27]. When NADPH oxidase is blocked, O_2_^−^ production decreases, NO bioavailability increases, oxidative stress decreases, and vascular remodeling is prevented [28]. THC also has the ability to reduce ROS by directly neutralizing and by enhancing the activity of endogenous antioxidant enzymes, such as SOD, CAT, glutathione peroxidase (GPx), and glutathione S-transferase (GST) [29,30]. It has been shown that polyphenols can regulate eNOS synthesis in vascular cells by activating transcription factors in the proximal parts of the eNOS promoter, such as Sp1, GATA, and others [31]. Hence, increased eNOS expression in vascular tissues in response to the treatment with THC could be a viable mechanism.

Reduced GSH and altered cellular redox status are caused by increased oxidative stress, as evidenced by significant increases in lipid peroxidation and protein oxidation in THC-treated *L-NAME* rats. THC, as previously mentioned, induces an increase in the expression of antioxidant enzymes followed by a decrease in the production of ROS (Figure 1). Just for comparison, THC was reported to be able to restore GSH levels to nearly 73%, while *L-NAME* caused GSH reduction of up to 59% from the control levels [32].

## 2. Protective Effects of THC in Cd^2+^-Induced Hypertension

Sangartit et al. [33] reported that THC protects against Cd^2+^-induced hypertension, raised arterial stiffness, and vascular remodeling in mice. Furthermore, it was also discovered that the changes in mechanical forces during Cd^2+^ exposure may result in the adaptive remodeling of the vascular walls [34].

The protein production and degradation processes cause a reorganization of the extracellular matrix, which is the crucial component of hypertensive vascular remodeling [34,35]. Recent studies have provided evidence of ROS-mediated activation of matrix metalloproteinase (MMP) by the involvement of NADPH oxidase as a consequence of increased mechanical elongation [36]. In the study of Sangartit et al. [33], mice exposed to Cd^2+^ had higher levels of arterial MMP-2 and MMP-9 expressions, linked to higher blood pressure and changes in vascular wall composition. Other studies [37,38] have shown that Cd^2+^ enhanced MMP-2 and MMP-9 levels followed by the induction of inflammation and proliferation. Enhanced MMP activation and reduced arterial compliance significantly contribute to hypertensive remodeling following Cd^2+^ exposure [38].

In the recent decade, there has been an increasing interest in the function of nutritional supplements in preventing and treating hypertension. In this direction, THC, as a potent antioxidant, has been proven to be antihypertensive by causing aortic stiffness to decrease [38]. On the other hand, the vascular endothelial cells can generate NO, which, despite antioxidant, anti-inflammatory, and antithrombotic activities, has a wide range of bioactivities, including smooth muscle relaxation; vasodilation; and suppression of cell growth, proliferation, and migration [39]. Reduced NO bioavailability can be induced by the suppression of the eNOS expression [40], a shortage of substrate or cofactors for eNOS [41], accelerated NO breakdown by ROS, and a shift in the cell signaling where eNOS is not effectively activated. In general, eNOS, as mentioned above, is the primary generator of NO in the vascular system. The recent findings demonstrate a substantial decrease in the aortic eNOS expression in Cd^2+^-exposed mice’s aorta. This is similar to the results of Yoopan et al., who demonstrated lower eNOS levels in the blood arteries of Cd^2+^-induced hypertensive rats [42].

Cd^2+^-induced vascular dysfunction compromises the integrity of the vascular endothelium and increases vascular inflammation [43]. Furthermore, tumor necrosis factor-α (TNF-α) and other pro-inflammatory mediators can stimulate inducible NOS (iNOS) activation. This is followed by NO-mediated dysregulation as a result of increased NO production [44] (Figure 2). Through an analysis of the iNOS activity, Chauhan et al. [45] showed that NO derived by iNOS provoked a reduction in eNOS expression and guanylate cyclase (GC) activity, causing endothelial and smooth muscle dysfunction. Overall, the findings of Sangartit et al. [33] revealed that THC restores the function and structure of the vasculature in Cd^2+^-induced hypertensive mice via the modulation of the eNOS/iNOS regulated pathway. In addition, Cd^2+^ possesses the ability to generate oxidative stress by inducing ROS and reactive nitrogen species (RNS) production, depleting GSH, inhibiting sulfhydryl (SH)-dependent enzymes by replacing some essential metals required for antioxidant enzyme activity, and/or increasing cell susceptibility to oxidative attack by altering membrane integrity and fatty acid composition [46,47]. Cd^2+^ exposure is linked to enhanced ROS formation (particularly O_2_^−^) [48]. Furthermore, increased O_2_^−^ production can trigger iNOS expression via increased NF-κB activity, which can enhance NO production [44]. As mentioned above, the induction of the iNOS results in an overabundance of NO, which interacts with O_2_^−^ and generates a severely potent oxidant: peroxynitrite (ONOO^−^) (Figure 2). Both O_2_^−^ and ONOO^−^ contribute to tissue damage and organ malfunction (including blood vessels, heart, liver, and kidney). THC administration lowers aortic O_2_^−^ generation and the urine nitrate/nitrite ratio in a dose-dependent manner. It was also shown that reduced iNOS expression is linked to lower O_2_^−^ levels and a lower nitrate/nitrite ratio, which inhibits ONOO^−^ production and increases NO bioavailability [49]. In addition, heavy metals, such as Cd^2+^, cause many adverse effects by generating free radicals, which result in DNA damage, lipid peroxidation, and a decrease in the protein sulfhydryl (e.g., GSH) [48,50]. THC significantly suppresses oxidant production, as evidenced by the prevention of high levels of O_2_^−^, MDA, and protein carbonyls, whereas the injection of THC into mice exposed to Cd^2+^ enhanced endogenous antioxidant GSH [48]. These findings imply that vascular protection in THC-treated animals is most likely attributable to oxidative stress regulation.

Additionally, THC was proven to have antihyperlipidemic and cardiovascular disease-preventive properties [10,51,52]. According to Pari and Amali [30], THC provides considerable protection against chloroquine toxicity by inhibiting lipid peroxidation via free radical scavenging, strengthening the antioxidant defense system. Cd^2+^ mainly accumulates in the liver, kidneys, heart, and aorta, causing tissue damage [53]. Sangartit et al. [33] discovered that when mice were exposed to Cd^2+^ accompanied by THC treatment, Cd^2+^ levels in these organs and the blood were lowered. The significant reduction in Cd^2+^ content in the blood following THC administration shows that THC, similar to CUR, can chelate Cd^2+^ by creating a metal–ligand complex, lowering Cd^2+^ overload in the body [54]. The chelating impact of THC is supported by significant improvements in hemodynamic and vascular responses; (even low doses of THC enhance these parameters) [54]. Furthermore, evidence from other trials demonstrated that THC had chelating and antioxidant capabilities in mice treated with Cd^2+^ directly by intravenous injection. In addition to being a chelating and antioxidant agent, THC may be able to interfere with gastrointestinal Cd^2+^ absorption, resulting in a decrease in Cd^2+^ concentration in the blood and tissues [54]. However, the precise mechanisms are unknown and warrant additional examination.

## 3. Vascular THC Protection in Conditions of Iron Overload

In a study focusing on the role of iron sucrose in the interactions between leukocytes and endothelium, Kuo et al. [55] found an index for early atherogenesis and subsequent atherosclerosis in a mouse remnant kidney model. Additionally, they discovered that iron treatment of human aortic endothelial cells elevated the expression of intracellular cell adhesion molecule-1 (ICAM-1), vascular cell adhesion molecule-1 (VCAM-1), and adhesion of U937 cells through upregulated NADPH oxidase and NF-κB signaling. In the same direction, [56] confirmed the link between inflammation and induction of iNOS, which is important for boosting NADPH oxidase activity via the c-Jun N-terminal kinase-AP1 (JNK-AP1) and Janus kinase 2/Interferon regulatory factor (Jak2/IRF) signaling pathways [57]. Thereby, it is proposed [58] that a small amount of NO generated by eNOS is required for cardiovascular homeostasis. In contrast, excessive amounts of NO produced by iNOS may harm the cardiovascular system [59]. Increased iNOS expression and activity have been linked to the pathophysiology of hypertension and its consequences [58]. The iron-induced over-regulation of iNOS causes a high NO production rate, which reacts with O_2_^−^ to generate ONOO^−^, resulting in nitrosative stress and endothelial dysfunction [60]. In iron-overloaded animals, oral treatment with deferiprone (L_1_) together with THC causes an increase in the endothelial NO production by up-regulating eNOS expression, enhancing endothelial function, and adequate baroreflex sensitivity. L_1_ and THC seriously reduced NADPH oxidase expression. The inhibition of the NADPH oxidase may help improve baroreflex sensitivity in iron-overloaded conditions [61]. In addition to antioxidants, L_1_ and THC can act as chelating agents by reducing iron deposits in iron-overloaded rats. THC has been found to have high antioxidant activity via the β-diketone moiety in the neutralization of O_2_^−^ and inhibiting oxidative stress [22]. Furthermore, THC-mediated free radical clearance may limit iNOS induction, resulting in decreased NO generation (Figure 3). THC can also cause iNOS reduction by inhibiting NF-κF-24]. THC restores endothelium protective properties by boosting eNOS and restoring vascular responses (61). In iron-overloaded mice, THC enhanced the antioxidant GSH and restored the redox status [61]. This might be related to the THC’s capacity to remove ROS mediators and boost GSH levels. Iron overload toxicity is commonly associated with free radical tissue damage. Sangartit et al. [61] also demonstrated that using either L_1_ or THC alone leads to the removal of the excess iron in the iron-overloaded condition. Furthermore, the combined therapy of L_1_ and THC causes non-transferrin-bound iron (NTBI) reduction in a condition of systemic iron overload (lowering blood pressure and improving baroreflex and vascular reactivity). The research of Thephinlap et al. [62] found that CUR and L_1_ therapy lowered plasma concentrations of non-transferrin-bound iron (NTBI) and malondialdehyde (MDA), and improved heart rate variability in rats with iron-induced iron overload. It has been proposed that the β-diketo moiety of CUR participated in NTBI chelation [63]. THC may have the same iron-chelating capabilities as CUR since it is a CUR analog with a β-diketone group that may be coupled with Fe (III) [64]. Although THC has a lower effect on iron chelation than L_1_, the synergistic effects of the two compounds in lowering iron overload could be a viable option in some severe iron overload circumstances.

## 4. THC-Mediated Mitochondrial Impact in Brain Vasculature

Decreased mitochondrial dynamics and malfunction play a significant role during brain ischemia. Mondal et al. [65] laid down fundamental knowledge about the new mechanism of mitochondrial failure during ischemia/reperfusion (I/R) injury via hyperhomocysteinemia (HHcy)-induced mitochondrial remodeling. The authors also implied that high homocysteine (Hcy) levels could indicate altered brain function during cerebral I/R. The same authors indicated that THC treatment decreased cerebral edema and cerebral blood flow and improved blood–brain barrier (BBB) damage following ischemic stroke [65]. BBB preservation is of utmost importance during an ischemic stroke [66]. THC maintains the integrity of the BBB by lowering endothelial cell damage [65]. Numerous studies have found total Hcy to be a significant, graded, and independent risk factor for coronary artery disease and stroke [67,68,69]. In addition, Mondal et al. [65] discovered that overall plasma and tissue Hcy levels are considerably higher in the ischemic groups compared to controls. THC treatment reduced Hcy levels in the ischemic brain. Furthermore, the same group discovered that ischemic stroke altered the expression of enzymes involved in Hcy metabolisms, such as cystathionine-β-synthase (CBS), cystathionine gamma-lyase (CSE), methylenetetrahydrofolate reductase (MTHFR), and S-adenosyl-L-homocysteine hydrolase (SAHH) (Figure 4). When the metabolism of cysteine or methionine is disrupted, Hcy levels in the body rise. Thus, the findings of Mondal et al. [65] demonstrate that I/R either directly interferes with methionine metabolism or indirectly changes methionine metabolic enzymes, which results in HHcy. In general, Hcy may be transported to the mitochondria during HHcy and cause hypermethylation by raising SAHH and reducing MTHFR levels (Figure 4). Additionally, Hcy was linked to cerebral arteriolar stiffness, endothelial damage, and, ultimately, brain dysfunction [70]. In the study of Mondal et al. [65], the levels of mitochondrial p47phox and gp91phox were elevated in the ischemic mitochondrial fraction. In contrast, thioredoxin reductase (TR) and manganese superoxide dismutase (MnSOD) levels were lowered, summarizing that the THC treatment improved these stress markers. Additionally, because CUR is an antioxidant [71], it has been proven that treating animals with CUR during I/R-induced injury ameliorates mitochondrial dysfunction by reducing oxidative stress. Cellular damage caused by neurological disorders or head injuries opens a transitory breach in the mitochondrial membrane, which can significantly limit adenosine triphosphate (ATP) production and even cause ATP hydrolysis [72]. Damage to brain tissue caused by I/R results in decreased glucose and oxygen transport, leading to much lower ATP synthesis. Mondal et al. [65] detected an increase in the brain’s mitochondrial permeability transition (MPT) pore as a consequence of ischemic stroke. This was followed by a subsequent decrease in O_2_ intake and ATP synthesis, whereas THC therapy alleviated these alterations. The authors also discovered a significant induction of matrix metalloproteinase (MMP-9) activity in the mitochondrial fraction isolated from the ischemic brain of I/R mice. Interestingly, THC treatment can drastically reduce MMP-9 activity [65]. MMP-9, a member of the MMP family that generally remodels the extracellular matrix (ECM), is overexpressed following cerebral ischemia, which has been linked to rapid matrix degradation, reduced BBB integrity, and increased infarct size after stroke [65,66]. Tissue inhibitors of metalloproteinase (TIMP-1 and TIMP-2) are endogenous MMP-9 inhibitors [73]. The balance between MMP and TIMP is vital for correctly functioning ECM remodeling and is required for various developmental and morphogenetic processes [74]. In this direction, Refsum et al. [75] reported a considerable rise in the expression of MMP-9 and a reduction in the expression of its tissue inhibitor TIMP-2. The increased MMP-9 protein/mRNA levels destroy tight junction proteins (TJPs) and improve BBB permeability [75]. TJPs are critical not just for tissue integrity, but also for vascular permeability, leukocyte extravasation, and angiogenesis [76]. TJPs control BBB function and keep MPT pores closed, providing them with neuroprotective characteristics. Mondal et al. [75] found that MMP-9 activation is related to reduced expression of cellular TJPs, zonula occludens-1 (ZO1), and occludin (in both protein and mRNA levels), which leads to mitophagy. These findings reveal that mitochondrial dysfunction in the ischemic brain downregulates ZO1 and occludin, mitigated by THC treatment. These findings imply that THC modifies the ischemic impact of cerebral vascular damage, perhaps by blocking MMPs/TIMPs, limiting TJP degradation, and conserving vascular integrity. It is generally understood that the prompt clearance of damaged mitochondria by autophagy (mitophagy) is essential for cellular homeostasis and function [77]. According to Mondal et al. [65], THC inhibits mitophagy in the mouse brain following damage induced by cerebral I/R. Thus, inhibiting mitophagy with THC treatment may aid in the reduction in ischemic damage. The same group discovered that DNA methyltransferases (DNMT1 and DNMT3a), enzymes responsible for DNA methylation and gene expression were drastically elevated in ischemic mitochondria. To further examine the mechanisms of THC-induced normalization of DNMT in ischemic mitochondria, Mondal et al. [65] found that the recovery DNMT levels were close to those of the control group (Figure 5). Furthermore, the S-Adenosylmethionine/S-Adenosylhomocysteine (SAM/SAH) ratio as a biomarker for clinical diagnosis of the atherosclerosis is much higher in the ischemic brain. More specifically, MMP-9 activity is elevated in ischemic mitochondria, resulting in a reduction in TIMP-2 regulation. TIMP-2 protein expression is considerably decreased in ischemia due to increased DNA hypermethylation and epigenetic inhibition of gene transcription [65]. Treatment with 5-Aza drastically changed TIMP-2 synthesis in the ischemic condition. These findings reveal that epigenetic DNA hypermethylation is triggered after ischemia reperfusion-induced damage, resulting in TIMP-2 suppression and ECM remodeling. The latter is one of the fundamental mechanisms that must be considered when investigating the THC-associated mechanisms related to ischemic conditions (Figure 5).

## 5. THC-Induced Mitochondrial Remodeling in Brain Vascular Endothelial Cells

Many neurodegenerative disorders, including ischemic stroke and Alzheimer’s disease, are associated with mitochondrial dysfunction [78,79,80]. As discussed above, some disorders have been linked to Hcy [81,82]. For these reasons, Vacek et al. [83] investigated the influence of THC on the control of mitochondrial dynamics in brain endothelial cells during HHcy. More specifically, they investigated how increased Hcy levels dysregulate mitochondrial fission and fusion equilibrium in mouse brain endothelial cells. Taking that endothelium regulates nutrition transport, it is reasonable to predict that damaged endothelium in the brain could substantially contribute to the pathophysiology of neurodegenerative disorders. According to Marchi et al. [80], detrimental stress for these cells is most typically associated with the increased generation of mitochondrial ROS. Furthermore, Vacek et al. [83] discovered a considerable increase in ROS production in Hcy-treated brain endothelial cells. THC therapy later showed a protective effect in Hcy-treated brain endothelial cells based on its free radical scavenging capability. Autophagy is frequently addressed in such research as a control process that balances the involved systems. Autophagy is a type of programmed cell death that is not apoptotic. Most evidence suggests that autophagy is primarily a pro-survival rather than a pro-death mechanism, at least in cells with intact apoptotic machinery [84]. Mitophagy, or selective mitochondrial autophagy, is a critical mitochondrial quality control process that relies on the presence of particular mitophagy regulators that guarantee selective mitochondrial sequestration [84]. In their study, Vacek et al. [83] found an Hcy-induced increase in the autophagy marker microtubule-associated protein light chain 3 (LC3) and a significant increase in its receptor expression, normalized by THC treatment.

Hollenbeck and Saxton [44] reported that the fusion and fission of these dynamic organelles are required for mitochondrial function. Proteins that control mitochondrial dynamics are linked to various cellular processes [85]. Mitochondrial fission and fusion are frequently viewed as a finely calibrated equilibrium in cells. However, they are not fully integrated, and quantitative knowledge of how these processes interact with other mitochondrial cellular processes is lacking. Mitochondrial fission and fusion play roles in mitochondrial integrity, electrical and biochemical binding, turnover, and DNA segregation and protection. Vacek et al. [83] concluded that higher fission events lead to more mitophagy events. THC, on the other hand, significantly inhibited such effects in the cells. Furthermore, fission and fusion machinery have been linked to programmed cell death pathways [86]. The previous study also revealed that Hcy induces endothelial cell death [86]. Vacek et al. [83] also discovered that Hcy significantly increased cell apoptosis, which was inhibited by THC administration. This can be linked to lower levels of total intracellular ROS, lending credence to the theory that mitophagy is caused by oxidative stress in the cell. Hence, it should be highlighted that THC is engaged in controlling mitochondrial fusion, and preliminary research in the field indicates that it has a favorable influence. An in-depth study is required to provide a better understanding of THC participation in the control of such systems.

## 6. Antiangiogenic and Anti-Hypoxic Properties of THC

Yoysungnoen et al. [87] found that in CaSki-xenografted cervical cancer nude mice, VEGF was overexpressed, and VEGF and microvascular density (MVD) were closely associated with cancerous tissues. These findings are consistent with the recent discovery that VEGF plays a crucial role in biological tumor activity and neovascularization [88]. In hypoxic conditions, HIF-1α regulates VEGF expression [89]. HIF-1α is a transcription factor that in mammals mediates cellular and systemic homeostatic responses to decreased O_2_ availability, such as angiogenesis, erythropoiesis, and glycolysis. During tumor growth, hypoxic zones are formed in the tumor mass, where HIF-1α can stimulate VEGF protein increase [90]. According to Yoysungnoen et al. [87], HIF-1α is considerably enhanced in CaSki-xenografted mice and hypoxia-induced angiogenesis is mediated by VEGF-induced diseased blood vessels. When VEGF binds to its high-affinity receptors (Flt-1, VEGFR-1, Flk-1, KDR, VEGFR-2), it initiates a signaling cascade that results in enhanced endothelial cell survival and proliferation, vascular permeability, cell migration, and invasion [91,92,93].

VEGF receptors VEGFR-1 and VEGFR-2 have similar structures, but different biological functions [92]. For example, whereas VEGFR-1 has little effect on endothelial cell proliferation [94], VEGFR-2 plays an essential role in activating downstream components involved in proliferation, such as endothelial cell invasion, migratory differentiation, and embryonic angiogenesis [95,96]. The PLC/PKC/MAPK signaling pathway is preferred by VEGFR-2 [97]. In addition, in CaSki-xenografted cervical cancer in nude mice, Yoysungnoen et al. [87] found a strong link between VEGFR-2 and VEGF. As a result, the VEGF-VEGFR-2 system is qualified as an important target for cancer antiangiogenic treatment. Due to the genetic stability, homogeneity, and low mutation rate of the endothelial cells, antiangiogenic therapy of malignancies offers the prospect of treatment with minimal toxicity without gaining drug resistance [98].

The Yoysungnoen et al.’s [87] research revealed that THC dosages of 100, 300, and 500 mg/kg lowered microvascular density. THC treatment also reduced pathogenic characteristics, such as host-induced microvascular dilatation, tumor-induced tortuosity, and permeability. This shows that THC medication is a method for modulating rather than eliminating the angiogenic process, resulting in vascular normalization. In their previous study, Yoysungnoen-Chintana et al. [88] demonstrated that in the CaSki mouse model, CUR was effective in inhibiting tumor development and angiogenesis at 1 g/kg [88]. The trial conducted later by Yoysungnoen et al. [87] determined that THC is effective at 100 mg/kg, which is ten times lower than the effective dose of CUR. THC may decrease HIF-1α expression and diminish VEGF and VEGFR-2 expressions during angiogenesis. This shows that THC is more efficient in suppressing angiogenesis than its parent chemical, CUR. THC is one of the principal in vivo metabolites of CUR and may play an essential role in the biological effects caused by CUR. THC, unlike CUR, is stable in phosphate buffer and at different pH levels [99]. Surprisingly, THC is readily absorbed via the gastrointestinal tract, suggesting that it is a favored prospective option as an anticancer therapy. THC significantly decreased microvascular density, HIF-1α, VEGF, and VEGFR-2 protein expression. On the one hand, the positive link between tumor promotion, development, and tortuosity, and the expression levels of HIF-1α, VEGF, and VEGFR-2, clearly indicates that THC suppresses tumor angiogenesis via the downregulation of the HIF-1α/VEGF/VEGFR-2 pathways [87].

## 7. Conclusions

The data obtained from experimental studies show that the active compounds in turmeric, CUR, and THC, alleviate the development of hypertension, improve the hemodynamic status, and restore vascular function in hypertensive NO-deficient rats. Further experimental data suggest that THC, as a chelating and antioxidant agent, plays a vital role in reducing the adverse vascular effects of Cd^2+^ and iron overload. In addition, THC, as one of the active anticancer forms of CUR, has been shown to significantly reduce cervical tumor angiogenesis and downregulate the HIF 1*α*/VEGF/VEGFR-2 pathway.

The protective properties of CUR and THC can be attributed to many factors, including the ability to neutralize ROS, increase NO bioavailability, and enhance the antioxidant GSH-dependent defense system. The potential of THC as a candidate for alternative treatment of cervical cancer in humans needs to be explored further. As a potent antioxidant, THC inhibits mitochondrial dysfunction and TIMP-2 hypermethylation in mice with ischemia reperfusion-induced injury. It has been established that THC possesses the potential to be used for preventive and therapeutic purposes in ischemia and has great physiological benefits.

In summary, further preclinical studies should be implemented to demonstrate THC’s vascular protective and anti-tumorigenic effects in humans. A significant benefit would be its preventive effect in the protection against brain stroke. Applying the methods used in the present reviewed studies would be useful and help define the doses and ways of THC administration in different disease settings.

## Figures and Tables

**Figure 1 molecules-27-05100-f001:**
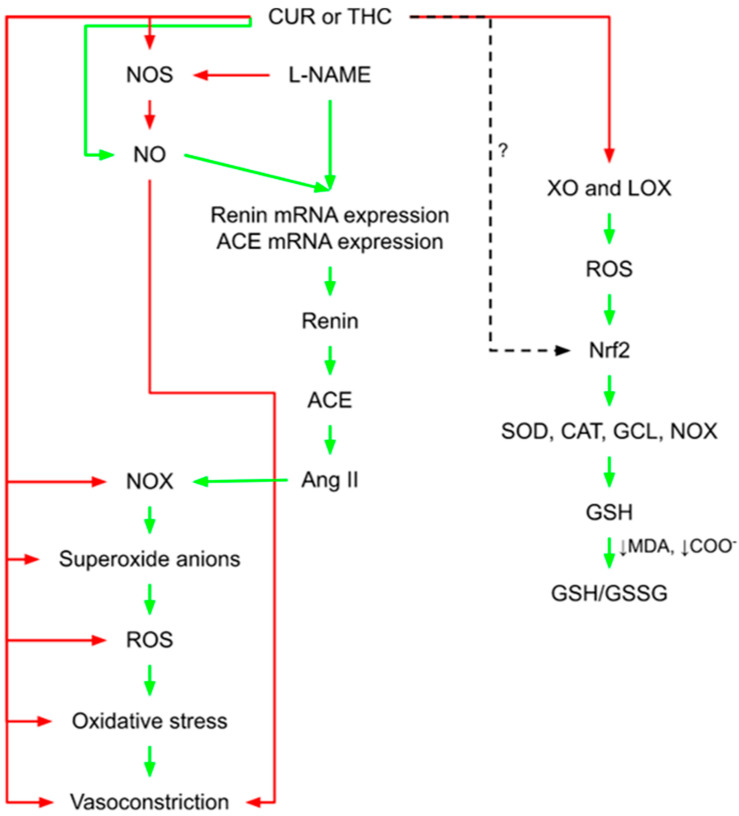
CUR- and THC-related vascular protection in L-NAME-induced hypertension and in the context of NO deficit. CUR activates Nrf2, increasing the expression of antioxidant enzyme genes, such as SOD, CAT, GCL, and NAD(P)H oxidoreductase. CUR and THC both reduce oxidant production by reducing the levels of O_2_^−^, MDA, and COO-, while endogenous GSH is boosted by raising the GSH ratio in the redox direction. THC does not appear to be able to activate Nrf2. CUR, curcumin; THC, tetrahydrocurcumin; L-NAME, (ω)-nitro-L-arginine methyl ester; Nrf2, nuclear factor erythroid-derived 2-like 2; SOD, superoxide dismutase; CAT, catalase; GCL, glutamate-cysteine ligase; NAD(P)H, oxidoreductase; O_2_^−^, superoxide radical; MDA, malondialdehyde; COO-, protein carbonyls; GSH, glutathione; punctuated line, unclear effect; red arrow, inhibitory effect; green arrow, stimulatory effect.

**Figure 2 molecules-27-05100-f002:**
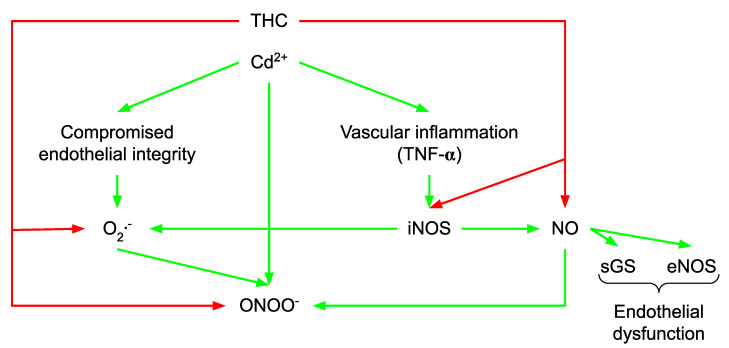
Protective effects of THC in Cd^2+^-induced hypertension. THC administration lowers Cd^2+^-induced aortic O_2_^−^ generation and the urine nitrate/nitrite ratio in a dose-dependent manner. Reduced iNOS expression is linked to lower O_2_^−^ levels and a lower nitrate/nitrite ratio, which inhibits ONOO^−^ production and increases NO bioavailability. THC, tetrahydrocurcumin; Cd^2+^, cadmium; O_2_^−^, superoxide radical; iNOS, inducible nitric oxide synthase; ONOO^−^, peroxynitrite; NO, nitric oxide; red arrow, inhibitory effect; green arrow, stimulatory effect.

**Figure 3 molecules-27-05100-f003:**
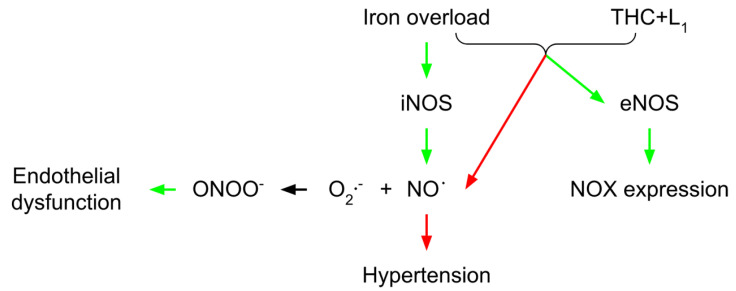
Vascular THC protection in conditions of iron overload. THC-mediated free radical clearance may limit iNOS induction, decreasing NO generation. THC restores endothelium protective properties by boosting eNOS and restoring vascular responses and blood pressure. THC enhanced antioxidant GSH and restored the redox status in iron-overloaded conditions. THC, tetrahydrocurcumin; iNOS, inducible nitric oxide synthase; eNOS, endothelial nitric oxide synthase; NO, nitric oxide; L_1_, deferiprone; red arrow, inhibitory effect; green arrow, stimulatory effect.

**Figure 4 molecules-27-05100-f004:**
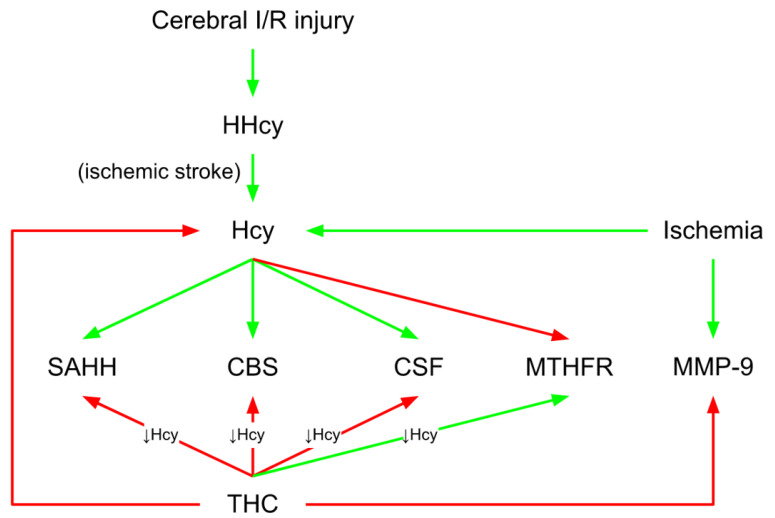
THC-mediated mitochondrial impact in brain vasculature. THC treatment reduced Hcy levels in the ischemic brain. Ischemic stroke altered the expression of enzymes involved in Hcy metabolisms, such as CBS, CSE, MTHFR, and SAHH. THC, tetrahydrocurcumin; Hcy, homocysteine; CBS, cystathionine-β-synthase; CSE, cystathionine gamma lyase; MTHFR, methylenetetrahydrofolate reductase; SAHH, S-adenosyl-L-homocysteine hydrolase; red arrow, inhibitory effect; green arrow, stimulatory effect.

**Figure 5 molecules-27-05100-f005:**
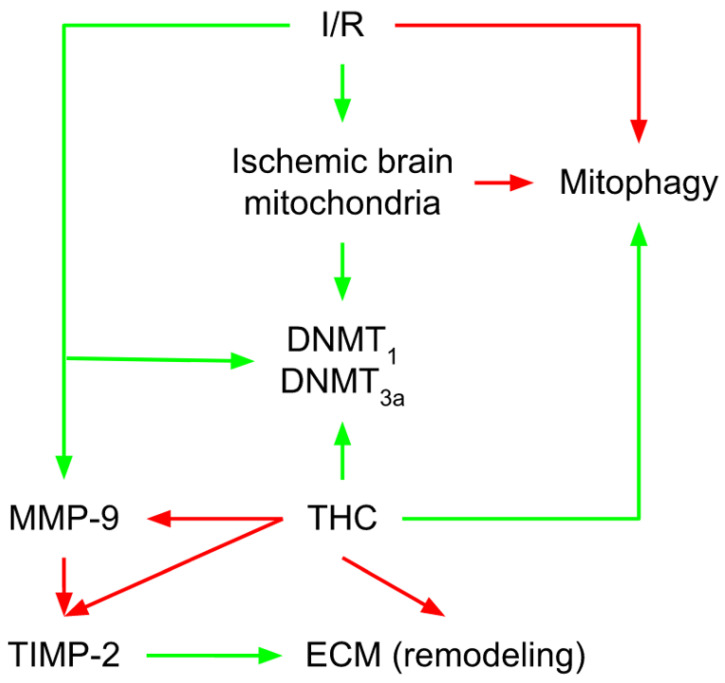
Mechanisms of THC-induced normalization of DNMT in ischemic mitochondria. THC-associated mechanisms related to ischemic conditions reveal that epigenetic DNA hypermethylation is triggered after ischemia reperfusion-induced damage, resulting in TIMP-2 suppression and ECM remodeling. THC, tetrahydrocurcumin; DNMT, DNA methyltransferase; TIMP-2, tissue inhibitor of metalloproteinase; ECM, extracellular matrix; red arrow, inhibitory effect; green arrow, stimulatory effect.

## Data Availability

Not applicable.

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
