# Peer review of "Tetrahydrocurcumin-Related Vascular Protection: An Overview of the Findings from Animal Disease Models"

_molecules, 2022, doi:10.3390/molecules27165100_

Round 1

Reviewer 1 Report

The work is well organized but because of the non upgrade references there is a question about the novelty for this work. I am suggesting a revision of the new data about the subject and after this major revision a new look on the material. 

Author Response

Dear Editor,

First of all, I would like to thank you very much for your attention and spent time in order to improve the quality of our work. Also, we appreciate the time and suggestions given by the reviewers and undoubtedly believe that by their implementation the quality of the manuscript will be significantly improved. For these reasons, I would like to point out that all suggestions and corrections of the manuscript are accepted and implemented in this new second version of the manuscript.

Answers to the Reviewer's suggestions, point by point:

Reviewer 1.

The work is well organized but because of the non-upgrade references, there is a question about the novelty of this work. I am suggesting a revision of the new data about the subject and after this significant revision a new look at the material. 

Answer: A revision of the new data was donned and information concerning cardiovascular effects of THC from the following references were additionally discussed and cited in the manuscript.

  1. Muhammad I, Muhammad N, Muhammad Asif K, et al. Curcumin and its allied analogs: epigenetic and health perspectives – a review. Czech J Food Sci. 2017;35(No. 4):285-310.
  2. A. Roth, C. Johnson, A. Abajobir, F. Abd-Allah, S.F. Abera, G. Abyu, et al., Global, regional, and national burden of cardiovascular diseases for 10 causes, 1990 to 2015, J. Am. Coll. Cardiol. 70 (1) (2017) 1–25.
  3. Cox FF, Misiou A, Vierkant A, et al. Protective effects of curcumin in cardiovascular diseases—impact on oxidative stress and mitochondria. Cells. 2022;11(3):342.
  4. Pourbagher-Shahri AM, Farkhondeh T, Ashrafizadeh M, Talebi M, Samargahndian S. Curcumin and cardiovascular diseases: Focus on cellular targets and cascades. Biomedicine & Pharmacotherapy. 2021;136:111214.
  5. Li KX, Wang ZC, Machuki JO, et al. Benefits of curcumin in the vasculature: a therapeutic candidate for vascular remodeling in arterial hypertension and pulmonary arterial hypertension? Front Physiol. 2022;13:848867.
  6. Alidadi M, Liberale L, Montecucco F, et al. Protective effects of curcumin on endothelium: an updated review. In: Guest PC, ed. Studies on Biomarkers and New Targets in Aging Research in Iran. Vol 1291. Springer International Publishing; 2021:103-119.
  7. Zhang ZB, Luo DD, Xie JH, et al. Curcumin’s metabolites, tetrahydrocurcumin and octahydrocurcumin, possess superior anti-inflammatory effects in vivo through suppression of tak1-nf-κb pathway. Front Pharmacol. 2018;9:1181.
  8. Rege SA, Varshneya MA, Momin SA. A Mini-Review: Comparison between curcumin and tetrahydrocurcumin based on their activities. Croat j food sci technol (Online). 2021;13(1):128-132.
  9. Lai CS, Ho CT, Pan MH. The cancer chemopreventive and therapeutic potential of tetrahydrocurcumin. Biomolecules. 2020;10(6):831.
  10. Kukongviriyapan U, Apaijit K, Kukongviriyapan V. Oxidative stress and cardiovascular dysfunction associated with cadmium exposure: beneficial effects of curcumin and tetrahydrocurcumin. Tohoku J Exp Med. 2016;239(1):25-38.
  11. Kamkin, G.A.; Kamkina, V.O.; Shim, I.A.; Bilichenko, A.; Mitrokhin, M.V.; Kazansky, F.V.; Filatova , S.T.; Abramochkin, V.D.; Mladenov, I.M. The role of activation of two different sGC binding sites by NO-dependent and NO-independent mechanisms in the regulation of SACs in rat ventricular cardiomyocytes. Rep. 2022, 10(7), e15246.
  12. Kuo, K.L.; Hung, S.C.; Lee, T.S.; Tarng, D.C. Iron sucrose accelerates early atherogenesis by increasing superoxide production and upregulating adhesion molecules in CKD. Am. Soc. Nephrol.2014, 25(11), 2596-2606.

Reviewer 2 Report

The manuscript "Tetrahydrocurcumin-related vascular protection: an overview of the findings from animal disease models" needs a moderate English change to improve the text, correction of misspelling, font format, and the number of items.

Line 87: "glutamate-cysteine ligase (GCL)" is written in a different font format;

Line 128: I strongly suggest that cadmium should be written as Cd2+ or Cd+2 instead of Cd. Please correct the whole manuscript if accept my suggestion.

Line 174: please format O2.- as shown in line 171.

Lines 194-210: format font and/or font size

Figure 3 legend: please insert the meaning of L1 in THC + L1.

Renumber/reorganize the items: 

1. Background (line 48)

2. Protective effects of THC in Cd-induced hypertension (line 125)

3. Vascular THC protection in conditions of iron overload (line 211)

4. THC-mediated mitochondrial impact in brain vasculature (line 263)

5. Antiangiogenic and anti-hypoxic properties of THC (line 398)

Conclusion section: try to conclude in one paragraph.

Author Response

Dear Editor,

First of all, I would like to thank you very much for your attention and spent time in order to improve the quality of our work. Also, we appreciate the time and suggestions given by the reviewers and undoubtedly believe that by their implementation the quality of the manuscript will be significantly improved. For these reasons, I would like to point out that all suggestions and corrections of the manuscript are accepted and implemented in this new second version of the manuscript.

Answers to the Reviewer's suggestions, point by point:

Reviewer 2.

The manuscript "Tetrahydrocurcumin-related vascular protection: an overview of the findings from animal disease models" needs a moderate English change to improve the text, correction of misspellings, font format, and the number of items.

Answer: The suggestion is accepted and proposed changes of language, misspelling, font, and number of items were carefully checked and corrected.

 Line 87: "glutamate-cysteine ligase (GCL)" is written in a different font format;

Answer: The font format is adequately changed.

Line 128: I strongly suggest that cadmium should be written as Cd2+ or Cd+2 instead of Cd. Please correct the whole manuscript if accept my suggestion.

Answer: The suggestion is accepted and proposed changes are introduced in the text throughout the whole manuscript. 

Line 174: please format O2.- as shown in line 171.

Answer: The format is adequately changed.

Lines 194-210: format font and/or font size

Answer: The font format and size were adequately changed

Figure 3 legend: please insert the meaning of L1 in THC + L1.

Answer: The meaning of L1 in the Fig. 3 legend is inserted

Renumber/reorganize the items: 

  1. Background (line 48)
  2. Protective effects of THC in Cd-induced hypertension (line 125)
  3. Vascular THC protection in conditions of iron overload (line 211)
  4. THC-mediated mitochondrial impact in brain vasculature (line 263)
  5. Antiangiogenic and anti-hypoxic properties of THC (line 398)

Answer: All listed items are adequately renumbered and reorganized.

Conclusion section: try to conclude in one paragraph.

Answer: The conclusion was adequately corrected.

Reviewer 3 Report

This paper reviewed the effects of tetrahydrocurcumin (THC) on the vascular protection in various animal disease models, and the underlying mechanisms were also discussed. This review is of importance to the understanding of potential roles of THC in vascular protection, and may lead to the further preclinical/clinical studies for the vascular-protective, antiangiogenic, and anti-tumorigenic effects of THC in humans. In general, the manuscript is well-organized but some key issues are required to be addressed before the publication on Molecules.

Major points:

1. In the background section, the recent review papers about the vascular protection effects of THC or its parent curcumin (CUR) should be discussed.

2. In the background section, the significance/meaning of this topic should be discussed, and the action of mechanism is suggested to be separated as an independent section with a subtitle for a better paper organization.

3. The authors discussed the effects of THC on vascular protection in multiple models. The effects of THC on iron sucrose-induced vascular dysfunction in mice should be included in the manuscript.

4. The highlights (findings from various disease models) should be strengthened for each section, and the discussion about the less related topics or background knowledge should be shortened in all the sections. Especially, under the title ‘THC-mediated mitochondrial impact in brain vasculature’, the description of MMP-9 and its relationship with TIMP-1/-2 as well as TJPs/ZO1, should be shortened. Under the title ‘THC-induced mitochondrial remodeling in the brain vascular endothelial cells’, the autophagy and mitophagy description should be shortened. Under the section ‘Antiangiogenic and anti-hypoxic properties of THC’, the discussion about the VEGF/VEGFR and cancers should be abbreviated significantly.

Minor points:

1. Description about the used different colors (red and green) in Figure legends should be included.

2. Artistry of Figures are suggested to be improved. For example, the authors can use different shapes with colors to illustrate the pathway instead of using only texts and arrows.   

3. The number ‘1’ at the beginning of almost all the titles makes confusions and should be removed.

Author Response

Dear Editor,

First of all, I would like to thank you very much for your attention and spent time in order to improve the quality of our work. Also, we appreciate the time and suggestions given by the reviewers and undoubtedly believe that by their implementation the quality of the manuscript will be significantly improved. For these reasons, I would like to point out that all suggestions and corrections of the manuscript are accepted and implemented in this new second version of the manuscript.

Answers to the Reviewer's suggestions, point by point:

Reviewer 3:

Comments and Suggestions for Authors

This paper reviewed the effects of tetrahydrocurcumin (THC) on vascular protection in various animal disease models, and the underlying mechanisms were also discussed. This review is of importance to the understanding of the potential roles of THC in vascular protection and may lead to further preclinical/clinical studies on the vascular-protective, antiangiogenic, and anti-tumorigenic effects of THC in humans. In general, the manuscript is well-organized but some key issues are required to be addressed before the publication on Molecules.

Major points:

  1. In the background section, the recent review papers about the vascular protection effects of THC or its parent curcumin (CUR) should be discussed.

Answer: A revision of the new data was donned and information concerning vascular protection effects of THC from the following references are additionally discussed in the manuscript.

  1. Muhammad I, Muhammad N, Muhammad Asif K, et al. Curcumin and its allied analogs: epigenetic and health perspectives – a review. Czech J Food Sci. 2017;35(No. 4):285-310.
  2. A. Roth, C. Johnson, A. Abajobir, F. Abd-Allah, S.F. Abera, G. Abyu, et al., Global, regional, and national burden of cardiovascular diseases for 10 causes, 1990 to 2015, J. Am. Coll. Cardiol. 70 (1) (2017) 1–25.
  3. Cox FF, Misiou A, Vierkant A, et al. Protective effects of curcumin in cardiovascular diseases—impact on oxidative stress and mitochondria. Cells. 2022;11(3):342.
  4. Pourbagher-Shahri AM, Farkhondeh T, Ashrafizadeh M, Talebi M, Samargahndian S. Curcumin and cardiovascular diseases: Focus on cellular targets and cascades. Biomedicine & Pharmacotherapy. 2021;136:111214.
  5. Li KX, Wang ZC, Machuki JO, et al. Benefits of curcumin in the vasculature: a therapeutic candidate for vascular remodeling in arterial hypertension and pulmonary arterial hypertension? Front Physiol. 2022;13:848867.
  6. Alidadi M, Liberale L, Montecucco F, et al. Protective effects of curcumin on endothelium: an updated review. In: Guest PC, ed. Studies on Biomarkers and New Targets in Aging Research in Iran. Vol 1291. Springer International Publishing; 2021:103-119.
  7. Zhang ZB, Luo DD, Xie JH, et al. Curcumin’s metabolites, tetrahydrocurcumin and octahydrocurcumin, possess superior anti-inflammatory effects in vivo through suppression of tak1-nf-κb pathway. Front Pharmacol. 2018;9:1181.
  8. Rege SA, Varshneya MA, Momin SA. A Mini-Review: Comparison between curcumin and tetrahydrocurcumin based on their activities. Croat j food sci technol (Online). 2021;13(1):128-132.
  9. Lai CS, Ho CT, Pan MH. The cancer chemopreventive and therapeutic potential of tetrahydrocurcumin. Biomolecules. 2020;10(6):831.
  10. Kukongviriyapan U, Apaijit K, Kukongviriyapan V. Oxidative stress and cardiovascular dysfunction associated with cadmium exposure: beneficial effects of curcumin and tetrahydrocurcumin. Tohoku J Exp Med. 2016;239(1):25-38.

  1. In the background section, the significance/meaning of this topic should be discussed, and the action of mechanism is suggested to be separated as an independent section with a subtitle for a better paper organization.

Answer: The suggestion is accepted and an independent section is introduced. Also based on their specificity the mechanisms of action were adequately discussed in every particular section. We are of the opinion that separation of the mechanisms could compromise some sections and the final picture will be even more complex.

  1. The authors discussed the effects of THC on vascular protection in multiple models. The effects of THC on iron sucrose-induced vascular dysfunction in mice should be included in the manuscript.

Answer: We agree with the reviewer's suggestion, and a completely new paragraph in the section “Vascular THC protection in conditions of iron overload“ was introduced. In this paragraph, we discussed the mechanisms affected by the iron sucrose-induced vascular dysfunction in mice. In addition, the following publication was introduced and discussed in the adequate section in the manuscript;

  1. Kuo, K.L.; Hung, S.C.; Lee, T.S.; Tarng, D.C. Iron sucrose accelerates early atherogenesis by increasing superoxide production and upregulating adhesion molecules in CKD. Am. Soc. Nephrol.2014, 25(11), 2596-2606.

  1. The highlights (findings from various disease models) should be strengthened for each section, and the discussion about the less related topics or background knowledge should be shortened in all the sections. Especially, under the title ‘THC-mediated mitochondrial impact in brain vasculature’, the description of MMP-9 and its relationship with TIMP-1/-2 as well as TJPs/ZO1, should be shortened. Under the title ‘THC-induced mitochondrial remodeling in the brain vascular endothelial cells, the autophagy and mitophagy description should be shortened. Under the section ‘Antiangiogenic and anti-hypoxic properties of THC’, the discussion about the VEGF/VEGFR and cancers should be abbreviated significantly.

Answer: All mentioned section contains only necessary descriptions concerning all mentioned processes. Having in mind the complexity of the above-mentioned processes we are on the opinion that a short introduction and reliable data could only help in a better understanding of the mechanisms and will lead the reader in the proper direction. Additional simplification of this part of the text could lead to misinterpretation of the manuscript, confusion of the reader, and loss of a clear picture of exactly what is going on. In addition, all indicated mechanisms are adequately cited.

Minor points:

  1. Description of the used different colors (red and green) in Figure legends should be included.

Answer: The required description is included in figure legends

  1. The artistry of Figures is suggested to be improved. For example, the authors can use different shapes with colors to illustrate the pathway instead of using only texts and arrows. 

Answer: The artistry of the Figures is improved.

  1. The number ‘1’ at the beginning of almost all the titles makes confusion and should be removed.

Answer: All items are properly renumbered and reorganized

Round 2

Reviewer 1 Report

I agree with this revised version.